# Impact of high-speed rail on tourism in China

Kehan Shi[1☯], Jinfang Wang[2☯], Xiaojin Liu[3]*, Xiaoying Zhao[4]

1 Institute of Industrial Economics of CASS, Beijing, China, 2 School of Economics and Management, Beijing Forestry University, Beijing, China, 3 School of Economics and Management, Jiangxi Agricultural University, Nanchang, Jiangxi, China, 4 College of Agriculture, Guangxi University, Nanning, China

☯ These authors contributed equally to this work.
* xiaojinjin06@163.com

## Abstract

The "time-space compression" effect of high-speed rail (HSR) has effectively improved the accessibility of the cities and has had a profound impact on tourism. This study explores the impact of HSR on tourism development in cities along HSR lines from the perspective of transfer of transport advantages, then evaluates the impact of HSR on tourism development using panel data of 286 cities in China from 2005 to 2013 by the difference-in-differences (DID) method. The empirical results show that the opening of HSR has significantly increased the tourism revenue and tourist arrivals. These results are still holds after considering endogenous HSR lines placement, and by various robustness checks. Further analysis of nodal effect shows that node cities experienced greater growth in tourism revenue than non-node cities. The analysis of mechanism found that tourism development in node cities relied on hotel industry, while tourism development in non-node cities relied on scenic spots industry. The findings of this study validate the role of HSR as a catalyst for urban tourism development, and reveal the comparative advantages of tourism in different cities under the influence of HSR. This study has important reference value for the development of tourism industry policies in cities along and around HSR lines.

**Data Availability Statement:** The data underlying the results presented in the study are available from National Bureau of Statistics of the People's Republic of China (http://www.stats.gov.cn/) and CRAD database of Chinese research data services platform (https://www.cnrds.com/).

## Introduction

Transportation infrastructure is an important connection between tourist sources and tourist destinations. As a representative of modern transportation infrastructure, the "time-space compression" effect of high-speed rail (HSR) weakens the impediment of spatial and temporal distance to the diffusion and concentration of tourism flows [1]. It provides a new impetus for tourism development. Since the first HSR in China (Beijing-Tianjin Intercity Railway) opened in 2008, China's HSR has developed rapidly. By the end of 2021, China had nearly 41,000 km of HSR in operation. Approach 93% of cities that population over 500,000 connected by HSR. It has become the main means of transportation for travelers [2]. In this context, Chinese local governments have attempted to use HSR to drive local tourism development. For example, regions along HSR, such as Sichuan, Jiangxi, and Yunnan, have successively established tourism alliances to jointly build tourism brands based on HSR connections to enhance regional tourism attraction.

**Funding:** The author(s) received no specific funding for this work.

**Competing interests:** The authors have declared that no competing interests exist.

At present, scholars have discussed a lot about the relationship between HSR and the development of tourism. Yang and Li [3] found that HSR significantly promoted the growth of the inbound tourism market. Zhang et al. [4] found that HSR promoted the growth of the domestic tourism market. Xin and Li [5] subdivided HSR into intercity HSR and non-intercity HSR, and found that intercity HSR could not promote the development of urban tourism, but could enhance the contribution of non-intercity HSR to urban tourism development. Wei et al. [6] verified the positive impact of HSR on urban tourism from the industrial efficiency. In addition, scholars have also found time lags [7] and persistence [8, 9] in the impact of HSR on urban tourism development. While most of the literature supports the view that HSR can promote the development of tourism, some scholars argue that this growth is only a level effect, not a rate effect, and suggest that only a small proportion of cities' tourism development can benefit from HSR construction [10].

The impact of HSR on urban tourism stems from the improvement of urban accessibility [11], which may be positive or negative. On the one hand, improved urban accessibility reduces travel resistance for tourists and releases significant tourism demand, but it also widens the travel time gap between tourists and destinations [12], so HSR will only have a positive impact on tourism if urban tourism appeal increases as a result of improved accessibility [13]. On the other hand, improved urban accessibility helps to weaken the geographical constraints on factor mobility, accelerating the concentration of tourism factors [14] and promoting the formation of tourism centres [15]. However, it has also been suggested that the relationship between tourism factor concentration and tourism development is not simply a positive one, and that excessive concentration of tourism enterprises and tourism labour can reduce the efficiency of the tourism industry [16]. As for the heterogeneity of the influence of HSR on the development of urban tourism, most of the existing literature attributed the reasons to the differences in urban socio-economic characteristics such as the geographical distribution [5], economic scale [17], resource endowment [18], city size [9], and level of administrative division [10], emphasizing the influence of Matthew effect, filtering effect, diffusion effect and superposition effect [19]. In fact, the impact of a city's tourism development from the HSR is closely related to the period of its construction. Existing literature shows that HSR mainly shows siphon effect in the formation period of main trunk line, and diffusion effect in the completion period of branch lines [20]. Kong and Li [21] also found that the strength of the promotion effect of HSR on urban tourism was related to whether the city was an original station or not.

The impact of HSR on tourism development is essentially the result of the adjustment of tourism resources configuration due to the change of transportation advantages [1, 15]. Chen et al. [22] elaborated this logic more clearly. They pointed out that the transportation network provides a supporting connection for urban connectivity and promotes cities to participate more in a wider range of factor flows relying on the "flow space". However, the "flow space" has nodes and centers, with cities having a hierarchy of influence [23]. The node level of cities in the HSR network determines its transportation advantage, and further determines its influence in the "flow space". Yin et al. [24] analyzed the influence of the improvement of HSR network on the tourism attraction of cities, and found that the greater the network density, the higher the tourism attraction of regions. Li et al. [25] confirmed the existence of node effect, but did not further discuss the mechanism of node effect.

To sum up, the existing research has conducted various discussions on the influence of HSR opening on tourism development. However, it has shown the following limitations: First, more literature analyzed the level effect of HSR on the economic growth of tourism, and rarely discussed the rate effect, so the contribution of HSR to tourism may be overestimated. Second, few literature discuss the reasons for heterogeneity of the impact of HSR on tourism development based on the differences in HSR factors, so it is difficult to explain the differences in the

role of HSR at different stages of construction. Third, the existing literature is mostly localized case studies on individual HSR lines, with the findings implying obvious regional characteristics, while possibly being influenced by other cross lines and poor generalization representation. In order to make up for the lack of available literature, this paper attempts to extend the existing research from the following aspects: Firstly, we use incremental values to measure changes in the tourism economy to examine the rate effect of HSR on tourism economic growth. Secondly, based on the point-axis theory, we constructed the mechanism of node effect, and divided cities into node and non-node cities based on the number of HSR lines passing through the city to test the node effect. Thirdly, we collated data on all HSR line in China since 2008 and conducted empirical tests on a sample of 286 cities in China to avoid possible regional limitations of the study findings and estimation bias caused by the omission of cross lines. The results show that HSR has a rate effect on tourism economic growth, but the effect strength is lower than the level effect estimated by the existing literature. Furthermore, the existence of the node effect makes different cities have different comparative advantages in the divided tourism industry. Our findings have good policy reference value for the local governments of HSR cities in promoting tourism development in the context of HSR network operations.

## Framework and hypothesis

### Time-space compression effect

Distance decay rate and destination accessibility are important factors affecting the implementation of tourism activities [26]. For tourists, the spatial distribution probability of tourism demand is related to the distance between the tourist destination and source. When the travel time becomes the main influencing factor of tourists' destination choice, the distribution probability gradually decreases with the distance [1]. For tourist destinations, the accessibility of the city is a prerequisite for ensuring that tourism activities can be carried out successfully. Fig 1 illustrates the impact of HSR on "time-space distances" between cities. When the spatial distance between cities is fixed, the opening of HSR shortens the commuting time between cities on the basis of improving the accessibility of cities (Fig 1A). When the commuting time to the destination city is fixed, the opening of HSR makes the spatial structure between cities more compact (Fig 1B). The emergence of HSR, on the one hand, weakened the negative impact of distance decay rate on tourism demand [27, 28], providing the possibility of transforming potential tourism demand into real tourism demand. On the other hand, it has enhanced the

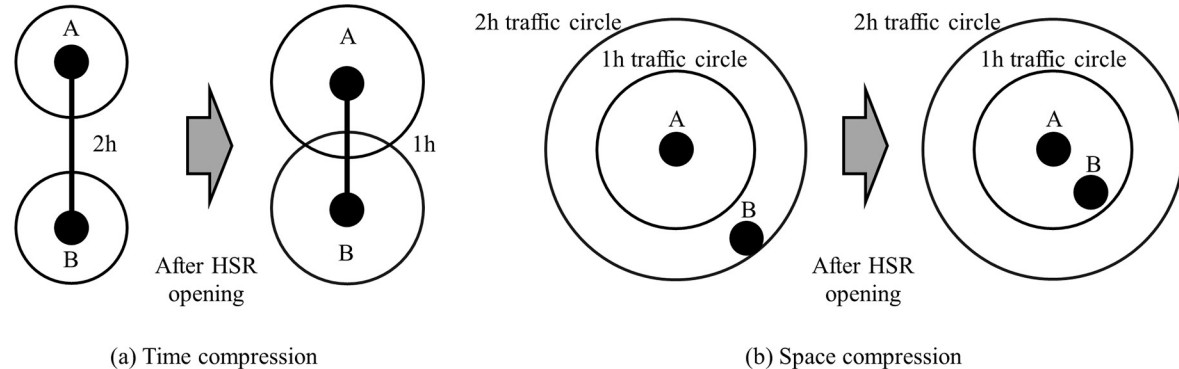

(a) Time compression

(b) Space compression

**Fig 1. The time-space compression effect of HSR.** (a) Time compression and (b) Space compression.

accessibility of cities along the route [29], which is conducive to expanding tourist source market [30] and increasing tourist attractiveness [31].

In addition, the HSR has accelerated the process of inter-regional tourism cooperation. The integration of tourism resources through regional cooperation is conducive to the creation of high-quality tourism products and itineraries and the formation of tourism economic corridors [32]. Tourists will also give priority to cities along HSR lines when choosing destinations to reduce the time and cost spent on transportation and maximize tourism utility [33]. Therefore, the following hypotheses are proposed:

H1: Given other conditions being equal, the opening of HSR is positively correlated with tourism revenue in cities along the route.

H2: Given other conditions being equal, the opening of HSR is positively correlated with tourist arrivals in cities along the route.

## Nodal effect

Based on the pole-axis theory, economic centres usually initially formed in a few well-located areas, and form a point-axis spatial structure by connecting with the surrounding areas through transportation routes [34]. Then industry sectors are clustered and growing in economic centres. After that, the product flow, capital flow, labor flow, technology flow, information flow, policy flow, etc. spreads to the surrounding areas through transportation routes, and regroups in the areas with sub-optimal locational conditions [15]. For tourism, the spatial distribution of tourism centres is also inextricably linked to transport factors [35]. Fig 2 draws the logical framework for the impact of HSR on urban tourism development based on the pole-axis theory.

The nodal effect of HSR reinforces the agglomeration effect of old tourism centres [36] and gives some cities that were less accessible the opportunity to develop into new tourism centres [37]. The nodal effect may cause heterogeneous development of urban tourism because of the different roles and benefits played by tourism centres and non-tourism centres in tourism activities [38, 39]. Node cities become distributing centres on travel routes, taking on a transit role, promoting tourism development by gathering more tourist flow and tourist consumption activities. Non-node cities become the radiating areas of the tourism centres. Their tourism development benefits from the increased local accessibility on the one hand [40], and the diffusion effect from the tourism centres on the other [41]. Furthermore, the differentiation between the tourism centre and the non-tourism centre is closely related to the development of the tourism subdivided industry [42]. The impact of HSR has led to differences in the

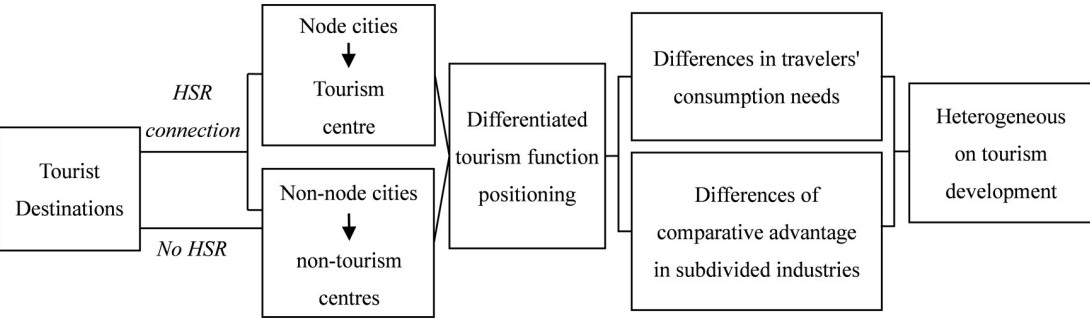

**Fig 2. The logical framework of the impact of HSR on urban tourism development.**

positioning of cities' tourism functions. The subdivided industries matching the cities' tourism function positioning forms a comparative advantage while influencing the tourism consumption structure, ultimately leading to heterogeneous tourism development in cities along the route. This gives rise to the following testable hypotheses:

H3: Given other conditions being equal, HSR has a greater positive impact on tourism revenue in node cities than in non-node cities.

H4: Given other conditions being equal, HSR has a greater positive impact on tourist arrivals in node cities than in non-node cities.

## Methods and data

### Econometric model

In recent years, the counterfactual framework is often used when assessing the treatment effect of a policy or event, comparing data that have been treated and assumed not to have been treated (counterfactual). DID is the most widely used measure to estimate the treatment effects of multiple treated objects. In fact, the timing of the opening of HSR (receiving treatment) is not consistent across cities. The assumption that all individuals in the treatment group of the traditional DID model receive treatment at the same time point cannot be satisfied. In this study, we follow Zhao et al. [43] design a multi-period DID model. The following benchmark model is set to identify the impact of HSR opening on urban tourism:

$$Tour_{it} = \theta HSR_{it} + \beta CONTROL_{it} + \mu_i + \lambda_t + \varepsilon_{it} \tag{1}$$

$$Tour_{it} = \theta_1 HSR_{it} \times Node_{it} + \theta_2 HSR_{it} + \beta CONTROL_{it} + \mu_i + \lambda_t + \varepsilon_{it} \tag{2}$$

Where subscripts $i$ and $t$ denote city and year, respectively. $Tour_{it}$ represents tourism development. $HSR_{it}$ is a dummy variable indicating whether the city has an HSR in operation. $Node_{it}$ denotes whether the city is a node city. $CONTROL_{it}$ represents a vector of control variables. $\mu_i$ and $\lambda_t$ indicate city fixed effects and time fixed effects. $\varepsilon_{it}$ is random disturbances. Firstly, we use Eq (1) to test the average effect of the impact of HSR on urban tourism development. Secondly, we use Eq (2) to capture the nodal effect of HSR impact on tourism. To avoid possible cross-sectional correlation, time-series correlation, and heteroscedasticity between cities, we use robust standard errors clustered at the city level. in all regressions.

### Variable selection

**Dependent variables.** In this study, we use two variables to measure tourism development (*Tour*), tourism revenue (*TR*) and tourist arrivals (*TA*). In the robustness testing section, dependent variables will be replaced by domestic tourism revenue (*DTR*) and domestic tourist arrivals (*DTA*). To test the rate effect of HSR on tourism development, all dependent variables use incremental data. For example, *TR* is calculated by subtracting the total tourism revenue in period *t-1* from the total tourism revenue in period *t*.

**Key independent variable.** The key independent variable we are interested in here, *HSR*, is a dummy variable that if the city opened HSR in period *t*, it takes the value of 1 in t and subsequent periods and 0 otherwise. Considering that there is a time lag in the impact of HSR on tourism development, we treat HSR that open in the first half of the year as opening in the current year and those that open in the second half as opening in the following year.

**Moderator variables.** To test the heterogeneity influence of HSR on tourism development caused by nodal effect, we construct a dummy variable (*Node*), which takes the value of 1 if the city with 2 or more HSR lines and 0 otherwise, as a moderating variable.

**Mechanism variables.** Differences in the positioning of tourism functions mean that the supply structure of tourism products differs from city to city. We focus on the two main categories of tourism product supply, including food and accommodation, and sightseeing.

1. Food and accommodation supply capacity (*FASC*).
   The number of hotels can reflect a city's capacity to meet the food and accommodation needs of tourists. Therefore, we select the number of star-rated hotels to measure *FASC*.

2. Sightseeing supply capacity (SSC).
   The ability of a city to meet the sightseeing needs of tourists is linked to the amount of high-quality tourism resources it possesses. Therefore, we select the number of China's 5A scenic spots to measure *SSC*.

**Control variables.** Drawing on existing research, we control for a range of city-level factors that may affect tourism development.

(1) Economic scale (*ES*).

The larger the city's economy, the better the foundation for tourism development. *TS* is measured by the natural logarithm of gross regional product.

(2) Fiscal expenditure (*FE*).

Higher fiscal expenditure means better urban infrastructure development, which helps to improve the tourism reception capacity of the city [8]. *FE* is measured by the natural logarithm of local government expenditure.

(3) Transportation convenience degree (*TCD*).

The cities with higher accessibility are more attractive to tourists, but it may have a negative impact on tourism revenue due to increased passenger mobility [5]. *TCD* is measured by the natural logarithm of highways passenger traffic.

(4) Industrial structure (*IS*).

Industrial structure change means the redistribution of production factors among different sectors, which affects the scale and the mode of economic growth, and in turn changes the disposable income and consumer awareness of potential source markets for tourism, ultimately affecting tourism development in terms of both the scale and structure of demand. We follow Wei et al. [6] use the industrial structure index to measure *IS*.

(5) Service industry support (*SIS*).

As one of the most interrelated service industries, the development of tourism depends on the support of other related industries. The scale of tertiary industry employees can reflect the overall supply capacity of the urban service industry. *SIS* is measured by the natural logarithm of the number of employed persons in the tertiary industry.

(6) Population size (*PS*).

Population size changes can influence tourism demand from local sources. Usually, the larger the population size, the higher demand for tourism from the local area [44]. *PS* is measured by the natural logarithm of the city's resident population.

(7) Income level (*IL*).

Income is the basis and prerequisite for consumption, and usually the higher the income, the higher the consumption. This means that cities with high income levels usually receive more tourism revenue [45]. *IL* is measured by the natural logarithm of the average wage of employed persons.

(8) Opening degree (*OD*).

The degree of opening facilitates the tourism industry to actively participate and share the information, technology, resources, and other valuable factors brought by internationalization. Usually, the higher degree of external openness, the higher the inbound tourism revenue of the region. *OD* is measured by the proportion of FDI in the GDP.

## Data description

China's first HSR construction started in 2005 and operated in 2008. To 2013, China's HSR operating mileage reached the total mileage of other countries combined. The main framework of China's HSR network has basically taken shape. After 2013, China's HSR networked layout has been further improved and cities have introduced industrial policies regarding "all-for-one tourism" strategies. To avoid the interference of these industrial policies in testing the impact of HSR on tourism development, the time window was set at a golden period of China's HSR development from 2005 to 2013. Data on tourism revenue and tourist arrivals are from China Statistics Yearbook for Regional Economy. Data on HSR opening times and HSR line information are from Chinese High-speed Rail and Airline Database (CRAD) in Chinese Research Data Services Platform (CNRDS). The data of HSR opening times and HSR lines in CRAD database are mainly from the construction and opening of HSR lines published by China State Railway Group Co., Ltd. since 2003. The rest of the city-level macroeconomic data are mainly sourced from the China City Statistical Yearbook, with some missing data supplemented from each city's Statistical Communique on National Economic and Social Development and each city's Statistical Yearbook. After removing the samples with serious data missing, we finally used the panel data of 286 cities, including four municipalities of Beijing, Tianjin, Shanghai, and Chongqing. The number of sample observations is 2574. The descriptive statistics of the above variables are shown in Table 1.

## Results

### Benchmark regression results

Table 2 presents the estimation results of the benchmark regression based on Eq (1). After controlling for both city fixed effects and time fixed effects, the coefficients on *HSR* were all significantly positive at the 1% statistical level (columns (1) and (3)). The direction and significance of the coefficients on *HSR* did not change after further inclusion of control variables (columns (2) and (4)). These results indicate that the opening of HSR has the rate effect on tourism revenue and tourist arrivals. In terms of economic significance, the opening of HSR increases tourism revenue increment and tourist arrivals increment in cities along HSR lines by 1.18% (0.313/26.420) and 9.03% (0.221/2.448) on average. H1 and H2 are proven.

The estimation results of the control variables are generally consistent with the established research. The economic scale of cities is positively correlated with tourist arrivals increment. The service industry support and population size of cities are positively correlated with tourism revenue increment. The degree of transportation convenience is negatively correlated with tourism revenue increment. Fiscal expenditure is negatively correlated with tourism revenue

**Table 1. Descriptive statistics of variables.**

| No. | Variable | Abbreviation | Observation | Mean | Std.Dev | Min | Med | Max |
|---|---|---|---|---|---|---|---|---|
| *Dependent variables* | | | | | | | | |
| 1 | Tourism revenue | TR | 2288 | 26.420 | 45.530 | -278.223 | 13.981 | 715.079 |
| 2 | Tourist arrivals | TA | 2288 | 2.448 | 4.155 | -14.476 | 1.489 | 93.244 |
| 3 | Domestic tourism revenue | DTR | 2288 | 25.486 | 43.211 | -256.390 | 13.535 | 609.100 |
| 4 | Domestic tourist arrivals | DTA | 2288 | 2.405 | 4.159 | -14.666 | 1.464 | 91.022 |
| *Independent variable* | | | | | | | | |
| 5 | HSR opening | HSR | 2574 | 0.155 | 0.362 | 0.000 | 0.000 | 1.000 |
| *Moderator variables* | | | | | | | | |
| 6 | Node level | Node | 2574 | 0.031 | 0.173 | 0.000 | 0.000 | 1.000 |
| *Mechanism variables* | | | | | | | | |
| 7 | Food and accommodation supply capacity | FASC | 2002 | 46.706 | 69.059 | 1 | 30 | 1168 |
| 8 | Sightseeing supply capacity | SSC | 2002 | 0.334 | 0.707 | 0 | 0 | 7 |
| *Control variables* | | | | | | | | |
| 9 | Economic scale | ES | 2574 | 0.197 | 0.517 | 0.000 | 0.000 | 5.000 |
| 10 | Fiscal expenditure | FE | 2574 | 15.889 | 0.993 | 13.014 | 15.822 | 19.191 |
| 11 | Transportation convenience degree | TCD | 2574 | 13.905 | 0.925 | 10.806 | 13.897 | 17.634 |
| 12 | Industrial structure | IS | 2574 | 8.630 | 0.998 | 4.382 | 8.608 | 12.566 |
| 13 | Service industry support | SIS | 2574 | 37.504 | 2.409 | 29.974 | 37.307 | 46.671 |
| 14 | Population size | PS | 2574 | 2.747 | 0.766 | -0.083 | 2.722 | 6.371 |
| 15 | Income level | IL | 2574 | 5.844 | 0.699 | 2.846 | 5.898 | 8.119 |
| 16 | Opening degree | OD | 2574 | 10.186 | 0.426 | 8.766 | 10.212 | 11.828 |

**Table 2. Benchmark regression results.**

| | TR | | TA | |
|---|---|---|---|---|
| | **(1)** | **(2)** | **(3)** | **(4)** |
| HSR | 0.364 *** (0.093) | 0.313 *** (0.079) | 0.259 *** (0.100) | 0.221 *** (0.082) |
| ES | | 0.273 (0.189) | | 0.538 ** (0.239) |
| FE | | -0.489 *** (0.144) | | -0.485 ** (0.221) |
| TCD | | -0.087 ** (0.039) | | -0.031 (0.033) |
| IS | | -0.055 (0.056) | | -0.046 (-0.053) |
| SIS | | 0.358 ** (0.167) | | 0.307 * (0.169) |
| PS | | 1.177 * (0.654) | | 0.728 (0.516) |
| IL | | -0.157 (0.136) | | -0.010 (0.117) |
| OD | | 0.012 (0.039) | | 0.044 (0.055) |
| Constant | -0.123 *** (0.016) | -3.134 (4.770) | -0.100 *** (0.168) | -6.618 (4.187) |
| R-square | 0.752 | 0.767 | 0.679 | 0.691 |
| Sample size | 2240 | 2181 | 2240 | 2181 |

Note

***, **, and* represent significance levels of 1%, 5%, and 10%, respectively; the values in parentheses are robust standard errors clustered at the city level; both city and year fixed effects are controlled in all the columns; *TR* = Tourism revenue; *TA* = Tourist arrivals; *ES* = Economic scale; *FE* = Fiscal expenditure; *TCD* = Transportation convenience degree; *IS* = Industrial structure; *SIS* = Service industry support; *PS* = Population size; *IL* = Income level; *OD* = Opening degree.

increment and tourist arrivals increment. Fiscal expenditure represents the degree of government intervention in the economy [5]. Although the original intention of government intervention is to play the macroeconomic control role of the government, it often produces excessive administrative intervention in the market and enterprises of tourism, which retards the basic role of the market in resource allocation and leads to the hindrance of the tourism economy. HSR, on the other hand, may further strengthen the administrative intervention of local governments in tourism.

## Parallel trend and dynamic effect

The average treatment effect of HSR affecting tourism development was analyzed in the previous section, but the year-on-year impact of HSR opening could not be observed from it. Moreover, the DID implies an important premise that tourism development in the sample cities should have parallel temporal trends if the HSR had not been in operation. Therefore, it is necessary to analyze the changes each year before and after the opening of HSR in the sample cities, which can observe the annual treatment effect of HSR opening on urban tourism development on the one hand, and the rationality of the identification strategy can be verified on the other hand.

Due to the inconsistent timing of the opening of HSR in each city. It is not possible to use the method of grouping to plot time trends to verify the changes in tourism development before and after the opening of HSR in cities along HSR lines versus non-HSR cities. Multi-period DID usually involves multiple periods before and after the treatment. Examining the treatment effects in each period before and after the treatment period can compare the differences before and after the treatment, which provides a new idea to test the parallel trend hypothesis in multi-period DID. We follow Beck et al. [46] to construct a multi-period DID model that includes the effects of each period before and after the opening of HSR to verify whether the parallel trend hypothesis is satisfied. The model is given as:

$$Tour_{it} = \sum_{\tau=1}^{m}\theta_{-\tau}HSR_{i,t-\tau} + \theta_0 HSR_{it} + \sum_{\tau=1}^{q}\theta_{\tau}HSR_{i,t+\tau} + \beta CONTROL_{it} + \mu_i + \lambda_t + \varepsilon_{it} \quad (3)$$

Where $\theta_{-\tau}$ denotes the treatment effect in $\tau$ periods before the opening of HSR; $\theta_\tau$ denotes the treatment effect in $\tau$ periods after the opening of HSR; $\theta_0$ denotes the treatment effect in the period of HSR opening. The rest of the variables have the same meaning as in Eq (1).

Fig 3A and 3B plots the average treatment effect of HSR on tourism revenue increment (*TR*) and tourist arrivals increment (*TA*) in periods before and after HSR opening and the 95% confidence interval of $\theta_{-\tau}$, $\theta_{+\tau}$, and $\theta_0$, respectively. As shown in Fig 2A and 2B, we can accept the null hypothesis of the consistent time-varying trends of *TR* and *TA* in both the HSR and non-HSR cities before the initiation of operations.

We can also find that the enhancement of *TR* in cities along HSR lines by opening of HSR is not only significant in the period of opening but also significant at a 5% level for 8 years after the opening (Fig 3A). The intensity of the impact will increase over time and eventually remain stable. This means that there is a temporal continuity in the impact of HSR on tourism revenue growth. The annual treatment effect of HSR opening on *TA* (Fig 3B) is different from that on *TR*. The opening of HSR significantly increases tourist arrivals in cities along HSR lines only in the short term (T≤t+5). This may be attributable to the fact that at the beginning of the opening of HSR, the cities along HSR lines had the transportation location advantage and the tourism attractiveness of the cities was greatly enhanced, and the number of tourist arrivals increased significantly. However, as the density of the HSR network increases, the highway transportation support between cities along HSR lines and non-HSR cities gradually improved. The co-opetition effect of HSR and other transportation begins to manifest. The number of tourists to non-HSR cities increases, and the transportation location advantage of

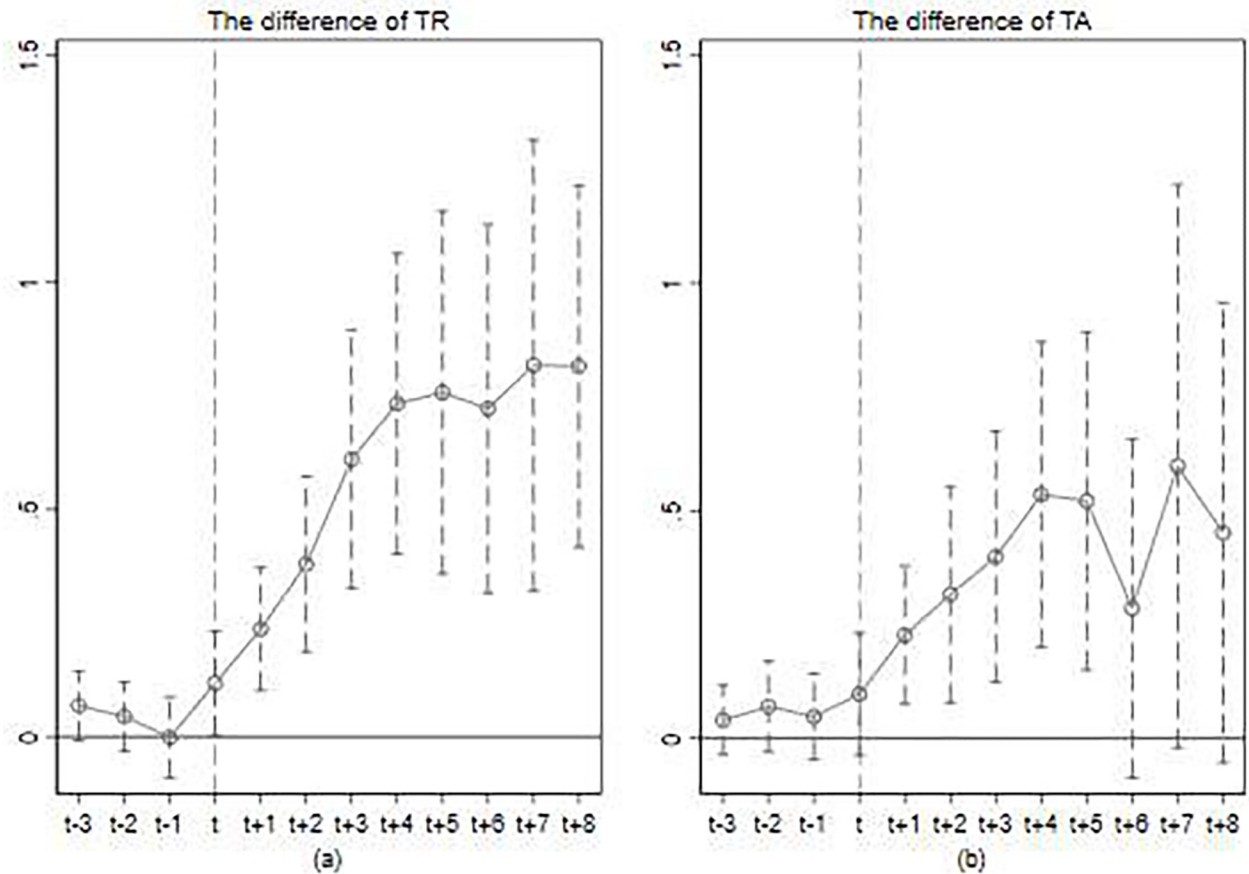

**Fig 3. Results of the parallel trend test for TR and TA.**

cities along HSR lines gradually disappears, which eventually leads to the promoting effect of the opening of HSR on the tourist arrivals becoming less significant in the long run.

## Results from the IV method

In addition to satisfying the assumption of parallel trends, the use of the DID to estimate the impact of HSR opening on tourism also requires the satisfaction that the opening of the HSR is exogenous, i.e., whether cities have HSR should be random and should not be disturbed by other measurable or unmeasurable factors that can affect tourism development. But this may not be the case. The main purpose of HSR construction is to reduce travel times between central or large cities, so economically and politically important cities are more likely to have HSR. In addition, omitted variables may affect both the opening of HSR and the development of tourism in the city, which would have led to biased estimates of the parameter estimates. For example, an official with a strong desire to develop local tourism may lobby hard for his city to become a HSR city; cities with advantageous tourism resource endowment are also more likely to be prioritized for HSR cities, etc. These uncontrollable and unobservable variables subject the HSR opening to endogeneity, which results in biased and inconsistent coefficient estimates of the variables and affects the accuracy of the estimation results in this study. We attempt to select appropriate instrumental variables to address the endogeneity of HSR opening in multi-period DID.

The select of instrumental variables needs to satisfy the conditions of both correlation and exogeneity, i.e., the instrumental variables should be correlated with the endogenous variables and uncorrelated with the random disturbance terms. The endogenous variable in this study is *HSR*, we follow Gao et al. [18] construct instrumental variable (*IVHSR*) by drawing straight lines between two end cities. The original intention of HSR construction is to connect central cities, so the straight lines we have drawn between end cities are closely related to the real HSR lines, i.e., *IVHSR* satisfies the condition correlated with *HSR*. However, whether a city is on one or more straight lines depends on geographical information and is therefore largely exogenous, i.e., *IVHSR* is uncorrelated with the random perturbation term in Eq (1), the exogeneity condition is also satisfied. Therefore, the regression model for the first stage of the instrumental variables approach corresponding to Eq (1) is:

$$HSR_{it} = \theta IVHSR_{it} + \beta CONTROL_{it} + \mu_i + \lambda_t + \varepsilon_{it} \tag{4}$$

Where *IVHSR*$_{it}$ is an instrumental variable of *HSR*$_{it}$ constructed with the straight-line strategy, and it takes the value of 1 when the city is located on straight lines between end cities in year *t* and 0 otherwise. Other variables and symbols have the same meaning as in Eq (1)

Table 3 reports the results of 2SLS estimations based on Eq (4). The results of the first stage show that the instrumental variable (*IVHSR*) is significantly and positively correlated with the endogenous variable (*HSR*) at a 1% level (column (1)). The F-statistic of 481.12 is much larger than the critical value of 10. These results indicate that the instrumental variable has a strong explanatory power for the endogenous variable. The results of the weak identification test show that the Cragg-Donald Wald statistic is much larger than the critical value corresponding to the tolerance of 10% distortion provided by Stock and Yogo [47], which indicates that *IVHSR* is not a weakly instrumental variable. The results of the underidentifiability test show that the Anderson LM statistic rejects the null hypothesis of "underidentifiable of instrumental variables" at a 1% level. Columns (2) and (3) report the results of the second stage estimation with *TR* and *TA* as dependent variables, respectively. It can be found that the coefficients of *HSR* on *TR* and *TA* are still significantly positive at a 1% level, and the coefficient values are

**Table 3. Results from the IV method.**

| | *HSR* | *TR* | *TA* |
|---|---|---|---|
| | (1) | (2) | (3) |
| *IVHSR* | 0.842 *** (0.038) | | |
| *HSR* | | 0.380 *** (0.096) | 0.273 *** (0.101) |
| Sample size | 2181 | 2181 | 2181 |
| Fist-stage F value | 481.12 | | |
| Weak identification test | 4426.104 | | |
| | <16.38> | | |
| Underidentification test | 117.668 | | |
| | [0.000] | | |

Note

***, **, and* represent significance levels of 1%, 5%, and 10%, respectively; the values in parentheses are robust standard errors clustered at the city level; results in column (1) is the first-stage estimation while those in columns (2) and (3) is second-stage estimation; *TR* = Tourism revenue; *TA* = Tourist arrivals; other controls include *ES* = Economic scale; *FE* = Fiscal expenditure; *TCD* = Transportation convenience degree; *IS* = Industrial structure; *SIS* = Service industry support; *PS* = Population size; *IL* = Income level; *OD* = Opening degree; both city and year fixed effects are controlled in all the columns.

larger compared to the estimated results in Table 2. It shows that after relieving the endogeneity of HSR lines placement, the opening of HSR still significantly boosts tourism development. The conclusions of this study remain unchanged.

## Robustness test

**Replacement of dependent variables.**   The dependent variables *TR* and *TA* used in benchmark regression both include the domestic tourism market and the inbound tourism market. In fact, the impact of HSR on tourism development is mainly focused on the domestic tourism market, while the inbound tourism market is more vulnerable to the airline system. Therefore, there may be some deviation in identifying the impact of HSR on tourism development using aggregate indicators. Accordingly, we re-estimated Eq (1) using domestic tourism revenue increment (*DTR*) and domestic tourist arrivals increment (*DTA*) as the dependent variables. The results show that the direction of coefficient effects and statistical significance of the main test variables remain consistent with the main regression results in Table 2 after replacing the dependent variables (columns (1) and (2) of Table 4).

**Eliminate the impact of national central cities.**   As mentioned earlier, HSR aims to connect central cities with important economic and political functions. Central cities have an advantage over neighboring cities in terms of access to resources, which makes them more conducive to tourism development [18]. The sample used in benchmark regression includes national central cities, which may affect the identification of the impact of HSR on tourism development. For this purpose, we re-estimated Eq (1) using the sample that excludes national central cities. The results show that the main empirical findings of this study have not changed (columns (3) and (4) of Table 4).

**Changing the time window of data.**   One may argue that estimation using data with shorter pre-processing periods gives more accurate results. The reason is that the longer the pre-processing period, the more likely the data will contain other noise shocks. During the period 2005–2013, China's tourism industry was affected by two shocks. The first was the formal proposal by the State Council in 2006 to make tourism a strategic pillar industry and the impact of tourism expanded across the board. The second was the gradual reform of the tourism market with the implementation of the Tourism Law of the People's Republic of China in 2013. To reduce these interferences, we re-estimated Eq (1) narrowing the time window to 2006–2012. The regression results show that the findings are still robust after shortening the pre-processing period of the data (columns (5) and (6) of Table 4).

**Table 4. Robustness checks with subsamples.**

| | Domestic tourism market | | Non-national central cities | | From 2006 to 2012 | |
|---|---|---|---|---|---|---|
| | *DTR* | *DTA* | *TR* | *TA* | *TR* | *TA* |
| | (1) | (2) | (3) | (4) | (5) | (6) |
| *HSR* | 0.320 *** (0.076) | 0.255 *** (0.087) | 0.298 *** (0.076) | 0.184 *** (0.068) | 0.334 *** (0.092) | 0.227 *** (0.088) |
| R-square | 0.758 | 0.651 | 0.806 | 0.686 | 0.758 | 0.715 |
| Sample size | 2181 | 2181 | 2141 | 2141 | 1626 | 1626 |

Note

\*\*\*, \*\*, and\* represent significance levels of 1%, 5%, and 10%, respectively; the values in parentheses are robust standard errors clustered at the city level; *TR* = Tourism revenue; *TA* = Tourist arrivals; *DTR* = Domestic tourism revenue; *DTA* = Domestic tourist arrivals; other controls include *ES* = Economic scale; *FE* = Fiscal expenditure; *TCD* = Transportation convenience degree; *IS* = Industrial structure; *SIS* = Service industry support; *PS* = Population size; *IL* = Income level; *OD* = Opening degree; both city and year fixed effects are controlled in all the columns.

## Heterogeneity analysis and mechanism identification

### Heterogeneity analysis

We estimated Eq (2) to analyse the heterogeneity impact of HSR on tourism caused by the nodal effect. The estimation results are presented in Table 5.

In Table 5, we find that the coefficient of $HSR^*Node$ is 0.588 and significant at a 1% level (column (1)). This indicates that the impact of HSR on tourism revenue is significantly different between node cities and non-node cities. It is easier for node cities to transform the location advantages brought by HSR into tourism industry development advantages. The coefficient of $HSR^*Node$ on tourist arrivals increment (*TA*) is not significant (column (2)). From the previous theoretical analysis, the opening of HSR increases the accessibility of non-node cities and strengthens the function of nodal cities as tourism distributing centres. Tourists visiting node cities may also visit the surrounding cities along HSR lines. Due to the superimposed effect, the growth of tourist arrivals in the cities along HSR lines may not be lower than that in node cities. Therefore, the nodal effect does not cause the heterogeneity impact of HSR on tourist arrivals. Comparing the results in columns (1) and (2) of Table 5 with columns (2) and (4) of Table 2, respectively. It can be observed that the coefficients of *HSR* are significantly smaller in all regressions after considering the nodal effect. The coefficient of $HSR^*Node$ is significantly positive at a 1% level in the regression of *TR* (column (1) of Table 5). This result shows that the nodal effect of the opening of HSR is mainly reflected in the rate effect on tourism revenue. In terms of economic significance, when cities along HSR lines become node cities, the HSR will increase the increment of tourism revenue by 0.588. This is equivalent to 187.86% of the average marginal effect of the opening of the HSR on tourism revenue growth (0.588/0.313). The above results support H3.

### Mechanism identification

The previous theoretical analysis shows that the opening of HSR forms new transportation network nodes, which causes the transfer of transportation location advantages, and eventually the transportation location advantages are transformed into tourism development advantages. Cities along HSR lines evolve into tourism centres and non-tourism centres, respectively. In this case, the positioning of tourism functions in each city will be different, and the tourism subdivided industry which matches the functional positioning will gain the comparative advantage. If HSR is used blindly for tourism development, it may lead to an irrational

**Table 5. Heterogeneity impact of HSR on tourism development.**

|  | *TR* | *TA* |
|---|---|---|
|  | (1) | (2) |
| *HSR* | 0.253 *** (0.066) | 0.200 *** (0.077) |
| $HSR^*Node$ | 0.588 *** (0.216) | 0.211 (0.180) |
| R-square | 0.780 | 0.693 |
| Sample size | 2181 | 2181 |

Note

***, **, and* represent significance levels of 1%, 5%, and 10%, respectively; the values in parentheses are robust standard errors clustered at the city level; *TR* = Tourism revenue; *TA* = Tourist arrivals; other controls include *ES* = Economic scale; *FE* = Fiscal expenditure; *TCD* = Transportation convenience degree; *IS* = Industrial structure; *SIS* = Service industry support; *PS* = Population size; *IL* = Income level; *OD* = Opening degree; both city and year fixed effects are controlled in all the columns.

**Table 6. The mechanism of HSR impact on tourism development.**

|  | Node cities | | Non-node cities | |
| --- | --- | --- | --- | --- |
|  | *TR* | *TA* | *TR* | *TA* |
|  | **(1)** | **(2)** | **(3)** | **(4)** |
| *FASC* | 3.692 (6.116) | 12.69 *** (3.498) | 0.149 (0.263) | -0.361 (0.229) |
| *SSC* | -0.329 (0.227) | -0.253 (0.175) | 0.144 *** (0.041) | 0.076 (0.051) |
| R-square | 0.996 | 0.996 | 0.949 | 0.884 |
| Sample size | 37 | 37 | 260 | 260 |

Note

***, **, and* represent significance levels of 1%, 5%, and 10%, respectively; the values in parentheses are robust standard errors clustered at the city level; *TR* = Tourism revenue; *TA* = Tourist arrivals; *FASC* = Food and accommodation supply capacity; *SSC* = Sightseeing supply capacity; other controls include *ES* = Economic scale; *FE* = Fiscal expenditure; *TCD* = Transportation convenience degree; *IS* = Industrial structure; *SIS* = Service industry support; *PS* = Population size; *IL* = Income level; *OD* = Opening degree; both city and year fixed effects are controlled in all the columns.

allocation of tourism production factors, which in turn may hinder tourism growth. According to the theoretical framework presented in the previous section, node cities are more likely to develop into tourism centres, while non-node cities and other cities without HSR will become non-tourism centres. From the tourism demand perspective, node cities with convenient transportation and good facilities are more appropriate as distributing centres and transit areas for tourists, while non-tourism centres with abundant tourism attractions are better able to meet tourists' needs for excursions and sightseeing. Therefore, tourism consumption in node cities may be dominated by accommodation, dining, shopping, and entertainment, while non-node cities will be dominated by sightseeing. To verify the above conjecture, we divide the samples into node and non-node cities. Then we perform group regression using the number of star-rated hotels measuring the capacity of the food and accommodation supply, and the number of China's 5A scenic spots measuring the capacity of the sightseeing supply. This section uses samples from 2007 onwards for the regressions because China's 5A scenic spots assessment was first carried out in 2007. The results are presented in Table 6.

The results in columns (1) and (3) of Table 6 show that the coefficient of *SSC* on *TR* is not significant in node cities, but significantly positive at a 1% level in non-node cities. The results in columns (2) and (4) of Table 6 show that the coefficient of *FASC* on *TA* is significantly positive at a 1% level in node cities, but not has a significant effect in non-node cities. The results show that the construction of hotel industry produces a significant boost to tourism development in node cities, while the construction of scenic spot industry produces a significant boost to tourism development in non-node cities and confirm the validity of the previous theoretical derivation. This also provides a theoretical basis and empirical evidence for how cities along the route can rationalize the layout of the tourism industry, as well as having important policy implications.

## Conclusions and discussion

Based on the panel data of 286 cities in China from 2005 to 2013, this study empirically analyzes the impact of HSR on tourism development and heterogeneity caused by the nodal effect by using the DID method. It is revealed that: (1) The impact of HSR opening on tourism revenue and tourist arrivals growth are significant positive. (2) There has a significant nodal effect of HSR on tourism development, and the positive effect of HSR on tourism revenue growth is

significantly higher in node cities than in non-node cities. (3) The results of the mechanism analysis show that the tourism development in node cities depends on hotel construction, while on scenic spot industry in non-node cities. These findings validate the theoretical hypotheses of this study. The opening of HSR not only improves the accessibility of non-node cities, but also strengthens the function of tourism distributing centres in node cities. Tourism revenue and tourist arrivals in both nodal and non-node cities are enhanced. The growth effect of node cities is higher than that of non-node cities in terms of tourism revenue, as tourism distributing centres may attract more tourism consumption. The results of the mechanism analysis provide evidence that the nodal effect leads to differences in the functional positioning of tourism in cities along HSR lines. As the influence of spatial and temporal distance on tourism activities gradually weakens, most tourists choose tourism centres (node cities) with convenient transportation and well-developed infrastructure as the base for accommodation and transit points throughout the journey. The hotel industry, which caters to the food and accommodation needs of tourists, has become an important driving force in supporting tourism development in such cities. However, non-tourism centres (non-node cities) tend to have high-quality tourism attractions that can better satisfy tourists' needs for sightseeing, so the scenic spot industry becomes an important driver to support tourism development in these cities.

As an academic research, this research has some innovative contributions. First, our study complements the findings reported by Zeng and Chen [8], Xin and Li [5], Feng and Cui [10]. Zeng and Chen [8] and Xin and Li [5] used the absolute value of tourism economic indicators to analyze the impact of HSR on tourism development only reflects the level effect rather than rate effect. Feng and Cui [10] performed logarithmic treatment on the tourism economic indicators, and the results reflected the percentage change of the urban tourism economic indicators by HSR. Our research uses the incremental value of tourism economic indicators in logarithmic form, which can better reflect the rate effect of HSR on tourism development and avoid the overestimation of the contribution of HSR to tourism development. The results obtained in this paper for the impact of HSR on tourism revenue tourist arrivals are indeed lower than those obtained by Zeng and Chen [8]. Second, this study focuses on the differences in HSR factors, which breaks the limitation of analyzing the heterogeneous impact of HSR on tourism from the perspective of urban socio-economic characteristics [9, 17, 18, 24, 36] verifies the existence of node effect in HSR network. Third, this study reveals the comparative advantages of tourism development between nodal cities and non-nodal cities under the influence of HSR, which provides new insight to explain the failure of HSR to improve tourism industry efficiency [6].

Based on the above research findings, this study makes the following policy recommendations from both enterprise and government dimensions, with a view to better exploiting the key role of HSR in driving the transformation of China's tourism industry towards high-quality growth. (1). When investing in cities along HSR lines, enterprises should consider the subdivided industry of tourism to which the investment project belongs and the node level of the city. According to the different needs of the cities for subdivided industry of tourism reasonable investment. It will not only enhance the rate effect of HSR on urban tourism with greater effectiveness, but also help to improve the productivity of enterprises and avoid the negative effects due to misallocation of resources. (2). The government should combine resource endowment and node level in a comprehensive judgment when laying out the tourism industry in the region. Node cities should first focus on the development of accommodation, catering, tourist transportation and travel agency industries to play the function of tourist distributing centres and strengthen the construction of tourism infrastructure services. Secondly, they also need to pay attention to the development of tourism shopping and cultural entertainment. Non-node cities should make tourism resource development and scenic spots

construction the key elements of tourism development. They also need to improve the quality of tourism products, enrich tourism products, and promote tourism consumption upgrading. In this way, not only can haphazard investment be avoided, but it is also conducive to the formation of comparative advantages of each city, which in turn will enhance the attractiveness of city's tourism.

This study innovatively discusses the nodal effect in the impact of HSR on tourism development from the perspective of transportation advantage transfer and analyzes the differentiated tourism function positioning of cities along HSR lines caused by the nodal effect. Even though this study has contributed to an improved understanding of the relationship between HSR and tourism development, it still has some limitations. The degree of impact of HSR on tourism flows is correlated with the degree of HSR network improvement. The impact of HSR on tourism development may have changed as the density of China's HSR network increased after 2013. However, the time window of this study is set to 2005–2013, which does not capture the changes in the impact of HSR on tourism development in the context of further improvement of HSR network after 2013, although it avoids mixing the effect of HSR with the effect of the post-2013 successive introduction of regional tourism industrial policies. This limitation is compensated to some extent by the robustness test. In 2013, China opened 12 new HSR lines, accounting for 44% of the total number of HSR lines from 2005 to 2012. It can be assumed that the difference between 2012 and 2013 in the degree of HSR network improvement is significant. If the effect of the degree of HSR network improvement on the HSR tourism effect is non-negligible, then the regression results using 2006–2012 in the robustness test should be significantly different from the benchmark regression results. However, the results are not. This suggests that the findings of this study are still informative.

## Supporting information

**S1 Table. Abbreviation comparison table.**
(DOCX)

## Author Contributions

**Conceptualization:** Kehan Shi, Xiaojin Liu.

**Data curation:** Kehan Shi, Xiaojin Liu, Xiaoying Zhao.

**Formal analysis:** Kehan Shi, Jinfang Wang.

**Investigation:** Kehan Shi, Xiaojin Liu, Xiaoying Zhao.

**Methodology:** Kehan Shi, Jinfang Wang.

**Project administration:** Xiaojin Liu.

**Resources:** Kehan Shi, Jinfang Wang.

**Software:** Kehan Shi, Jinfang Wang.

**Supervision:** Xiaojin Liu.

**Validation:** Kehan Shi, Jinfang Wang, Xiaojin Liu.

**Visualization:** Kehan Shi, Jinfang Wang, Xiaojin Liu.

**Writing – original draft:** Kehan Shi, Xiaojin Liu, Xiaoying Zhao.

**Writing – review & editing:** Kehan Shi, Jinfang Wang, Xiaojin Liu.

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
