## [Decision Letter · Decision Letter 0]

29 Jul 2022

PONE-D-22-18625Impact of high-speed rail on tourism in ChinaPLOS ONE

Dear Dr. Liu, Thank you for submitting your manuscript to PLOS ONE. After careful consideration, we feel that it has merit but does not fully meet PLOS ONE’s publication criteria as it currently stands. Therefore, we invite you to submit a revised version of the manuscript that addresses the points raised during the review process. Please submit your revised manuscript by Sep 12 2022 11:59PM. If you will need more time than this to complete your revisions, please reply to this message or contact the journal office at plosone@plos.org. Please include the following items when submitting your revised manuscript:A rebuttal letter that responds to each point raised by the academic editor and reviewer(s). You should upload this letter as a separate file labeled 'Response to Reviewers'.A marked-up copy of your manuscript that highlights changes made to the original version. You should upload this as a separate file labeled 'Revised Manuscript with Track Changes'.An unmarked version of your revised paper without tracked changes. You should upload this as a separate file labeled 'Manuscript'.

We look forward to receiving your revised manuscript.

Kind regards,

Jun Yang

Academic Editor

PLOS ONE

Journal Requirements:

2. PLOS ONE does not copy edit accepted manuscripts (https://journals.plos.org/plosone/s/criteria-for-publication#loc-5). To that effect, please ensure that your submission is free of typos and grammatical errors.

Additional Editor Comments:

Major Revision

Reviewers' comments:

Reviewer's Responses to Questions

**Comments to the Author**

1. Is the manuscript technically sound, and do the data support the conclusions?

Reviewer #1: Yes

Reviewer #2: Partly

2. Has the statistical analysis been performed appropriately and rigorously? 

Reviewer #1: Yes

Reviewer #2: Yes

3. Have the authors made all data underlying the findings in their manuscript fully available?

Reviewer #1: Yes

Reviewer #2: Yes

4. Is the manuscript presented in an intelligible fashion and written in standard English?

Reviewer #1: Yes

Reviewer #2: Yes

5. Review Comments to the Author

Reviewer #1: The authors explored the impact of high-speed rail on tourism in China. The research methodologies are reasonable, and the findings are interesting. However, there are still some aspects that should be improved to make the paper publishable. I focus here only on some points, which are hopefully easy for the authors to take into account in the revision.

1. Part Abstract - please highlight the contribution of the research.

2. Part Introduction - Line 51, full name of HSR should be given when it first appeared. The innovation should be highlighted. There are some references on this topic, I suggest you supplied it in this part, as follows.

(1) The influence of high-speed rail on ice–snow tourism in northeastern China. Tourism Management (2020), doi:10.1016/j.tourman.2019.104070.

(2) Effects of rural revitalization on rural tourism. Journal of Hospitality and Tourism Management (2021),https://doi.org/10.1016/j.jhtm.2021.02.008.

3. Table 1 - negative number, such as minimum of TR and TA?

4. There are many mistakes, such as Line 92 lev-els. Please modify it carefullt.

5. The authors chosen 2005 to 2013 as the study period. Why not choose recent year?

6. Part Discussion looks very common, comparison with previous studies should be given.

Reviewer #2: The manuscript titled " Impact of high-speed rail on tourism in China" explored the impact of high-speed rail (HSR) on tourism development in cities along HSR lines from the perspective of transfer of transport advantages, then evaluates the impact of HSR on tourism development using panel data of 286 cities in China from 2005 to 2013 by the difference-in-differences (DID) method. Overall, the research is of potential good quality and interest to the journal readership. However, several parts need to be further improved:

(1) Section 1. In the introduction the logic is not fluent and the innovation is not outstanding.

(2) The literature section should be added to highlight the shortcomings of existing research and the innovation of this research.

(3) there are some grammatical problems throughout the whole manuscript. Therefore, the writing must be improved. For example, line 141. “The nodal effect of HSR reinforces the agglomeration effect of old tourism centres [1] and given some cities that were less accessible the opportunity to develop into new tourism centres.”, given or gives?

(4) Line 207-244. It is recommended that control variables be represented in tables.

(5) There are too many acronyms in the text. They could be summarized in a table either at the beginning or end of the article.

(6) Line 245-259. Although the data source were explained, the purpose and authoritative interpretation of the study data were not enough.

(7) Line 260. The expression of Table 1 should be adjusted and a three-line table is recommended.

(8) Line 283, Line 371, Line 406, Line 415, Line 457. The expression of Table 2,Table 3, Table 4, Table 5 and Table 6 should be adjusted and a three-line table is recommended.

(9) The discussion section should be re-written to underline the importance and relevance of control variables, and the implication (e.g. practical and academic implications) should be also clearly stated.

6. PLOS authors have the option to publish the peer review history of their article (what does this mean?). If published, this will include your full peer review and any attached files.

Reviewer #1: No

Reviewer #2: No

---

## [Author Response · Author response to Decision Letter 0]

21 Aug 2022

Dear editor:

Thank you for your kind letters of “PLOS ONE Decision: Revision required [PONE-D-22-18625R1]” on 29-Jul-2022, and for the reviewers’ comments concerning our manuscript entitled “Impact of high-speed rail on tourism in China”. We revised the manuscript in accordance with the editor’s and reviewers’ comments, and carefully proof-read the manuscript to minimize typographical, grammatical, and bibliographical errors.

Here below is our description on revision according to the reviewers’ comments. (The review comments are in black and our response notes are in purple)

Part A (Reviewer 1)

The authors explored the impact of high-speed rail on tourism in China. The research methodologies are reasonable, and the findings are interesting. However, there are still some aspects that should be improved to make the paper publishable. I focus here only on some points, which are hopefully easy for the authors to take into account in the revision.

1. Part Abstract - please highlight the contribution of the research.

The authors’ Answer: Thank you for your kind comment. As suggested, we have added the following to the abstract to highlight the contribution of this research: “The findings of this study validate the role of HSR as a catalyst for urban tourism development and reveal the comparative advantages of tourism in different cities under the influence of HSR. This study has important reference value for the development of tourism industry policies in cities along and around HSR lines.” (Line 45-48).

2. Part Introduction - Line 51, full name of HSR should be given when it first appeared. The innovation should be highlighted. There are some references on this topic, I suggest you supplied it in this part, as follows.

(1) The influence of high-speed rail on ice–snow tourism in northeastern China. Tourism Management (2020), doi:10.1016/j.tourman.2019.104070.

(2) Effects of rural revitalization on rural tourism. Journal of Hospitality and Tourism Management (2021),https://doi.org/10.1016/j.jhtm.2021.02.008.

The authors’ Answer: Thank you for your kind comment. We have revised the paper as suggested.

Firstly, we have added the full name of HSR at where it first appeared (Line 54).

Secondly, we have rewrote the introduction and summarize the limitations of the existing literature (Line 103-108), then we propose Innovative contribution of this research: (1) We use incremental values to measure changes in the tourism economy to examine the rate effect of HSR on tourism economic growth. (2) Based on the point-axis theory, we constructed the mechanism of node effect, and divided cities into node and non-node cities based on the number of HSR lines passing through the city to test node effect (Line 108-114).

Thirdly, we have cited these two articles you provided. We consider that these two references are not only useful in furthering our understanding of this topics in this paper, but also have important implications for the construction of the theoretical framework of this paper (Line 74 and 156).

3. Table 1 - negative number, such as minimum of TR and TA?

The authors’ Answer: Thank you for your kind comment. Table 1 reports the results of descriptive statistics for each variable in the model. The minimum values of TR, TA, DTA and DTR have negative number. This is because we have log-transformed the raw data for these variables (the calculation method of these variables is introduced in the part of Variable selection), and Table 1 reports the log-transformed variable values. The log transformation can be used to make highly skewed distributions less skewed. This can be valuable both for making patterns in the data more interpretable and for helping to meet the assumptions of inferential statistics.

4. There are many mistakes, such as Line 92 lev-els. Please modify it carefullt.

The authors’ Answer: Thank you for your kind comment. We re-checked the manuscript and modified any mistakes in the use of uninterrupted hyphens.

5. The authors chosen 2005 to 2013 as the study period. Why not choose recent year?

The authors’ Answer: Thank you for your kind comment. Between 2008 and 2012, the concept of “all-for-one tourism” was introduced by individual cities in China. Since 2013, more and more cities have started to implement “all-for-one tourism” strategie and introduced various tourism industry policies. To avoid the interference of these industrial policies in test the impact of HSR on tourism development, the time window was set at 2005-2013. However, this study period could not capture the changes in the impact of HSR on tourism development in the context of further improvement of HSR network after 2013. This limitation is compensated to some extent by the robustness test. In 2013, China opened 12 new HSR lines, accounting for 44% of the total number of HSR lines from 2005 to 2012. It can be assumed that the difference be-tween 2012 and 2013 in the degree of HSR network improvement is significant. If the effect of the degree of HSR network improvement on the HSR tourism effect is non-negligible, then the regression results using 2006-2012 in the robustness test should be significantly different from the benchmark regression results. However, the results are not. This suggests that the findings of this study are still informative.

6. Part Discussion looks very common, comparison with previous studies should be given.

The authors’ Answer: Thank you for your kind comment. We have added these in the part of conclusions and discussion. Specifically, we compare our findings with the existing literature around the innovations proposed in the part of introduction (Line 509-521).

Part B (Reviewer 2)

The manuscript titled " Impact of high-speed rail on tourism in China" explored the impact of high-speed rail (HSR) on tourism development in cities along HSR lines from the perspective of transfer of transport advantages, then evaluates the impact of HSR on tourism development using panel data of 286 cities in China from 2005 to 2013 by the difference-in-differences (DID) method. Overall, the research is of potential good quality and interest to the journal readership. However, several parts need to be further improved:

1. Section 1. In the introduction the logic is not fluent and the innovation is not outstanding.

The authors’ Answer: Thank you for your kind comment. We have reorganized the logic of the introduction and rewrote it. In the modified introduction, we first introduce the research background. Second, we reviewed the existing literature from three aspects: the relationship between HSR and tourism development, the heterogeneous impact of HSR on tourism and the causes of this heterogeneity. Third, we summarize the limitations of the existing literature and propose the innovative contribution of this paper (Line 62-114).

2. The literature section should be added to highlight the shortcomings of existing research and the innovation of this research.

The authors’ Answer: Thank you for your kind comment. We added the literature section to the part of introduction as suggested. Our literature review mainly includes the following three aspects: the relationship between HSR and tourism development, the heterogeneous impact of HSR on tourism and the causes for this heterogeneity (Line 63-102). Through literature review, we summarized the limitations of the existing literature (Line 103-108) and put forward our innovative contributions (Line 108-114).

3. There are some grammatical problems throughout the whole manuscript. Therefore, the writing must be improved. For example, line 141. “The nodal effect of HSR reinforces the agglomeration effect of old tourism centres [1] and given some cities that were less accessible the opportunity to develop into new tourism centres.”, given or gives?

The authors’ Answer: Thank you for your kind comment. We revised the manuscript in accordance with reviewers’ comments, and carefully proof-read the manuscript to minimize grammatical errors.

4. Line 207-244. It is recommended that control variables be represented in tables.

The authors’ Answer: Thank you for your kind comment. The number of control variables chosen for this paper is large, including Economic scale (ES), Fiscal expenditure (FE), Transportation convenience degree (TCD), Industrial structure (IS), Service industry support (SIS), Population size (PS), Income level (IL) and Opening degree (OD). Listing them one by one may not be conducive to the reader getting the key information quickly. Reporting control variables in tables can improve the reader's reading efficiency. However, in the part of control variables we not only introduced the variable symbols, the meaning of the variables and how the variables are calculated, but also explained why these variables were chosen as control variables. It is difficult to present all of these in a concise table. After weighing this, we have decided to keep the current presentation of control variables and added a column in Table 1 to indicate the categories of variable (Line 274), so that the reader can more intuitively understand what the control variables are.

5. There are too many acronyms in the text. They could be summarized in a table either at the beginning or end of the article.

The authors’ Answer: Thank you for your kind comment. We used a lot of variable acronyms in analyzing the regression results, which has caused some difficulty for the reader in understanding the results. To make the meaning of each acronym clearer to the reader, we have added a column of the meaning of acronyms in Table 1 (Line 274).

6. Line 245-259. Although the data source were explained, the purpose and authoritative interpretation of the study data were not enough.

The authors’ Answer: Thank you for your kind comment. We have improved this section as suggested (Line 262-270). The data we used in this paper includes two types, one is the macroeconomic data at the city level, and the other is the data on HSR lines. The macroeconomic data are from China Statistics Yearbook for Regional Economy, China City Statistical Yearbook, each city’s Statistical Communique on National Economic and Social Development, and each city’s Statistical Yearbook. The data on HSR lines are from Chinese High-speed Rail and Airline Database (CRAD) in Chinese Research Data Services Platform (CNRDS). The data of HSR opening times and HSR lines in CRAD database are mainly from the construction and opening of HSR lines published by China State Railway Group Co., Ltd. since 2003, which is highly authoritative.

7. Line 260. The expression of Table 1 should be adjusted and a three-line table is recommended.

The authors’ Answer: Thank you for your kind comment. We have modified the format of Table 1 according to the Table Guidelines of the journal.

8. Line 283, Line 371, Line 406, Line 415, Line 457. The expression of Table 2,Table 3, Table 4, Table 5 and Table 6 should be adjusted and a three-line table is recommended.

The authors’ Answer: Thank you for your kind comment. We have modified the format of Table 2, Table 3, Table 4, Table 5, and Table 6 according to the Table Guidelines of the journal.

9. The discussion section should be re-written to underline the importance and relevance of control variables, and the implication (e.g. practical and academic implications) should be also clearly stated.

The authors’ Answer: Thank you for your kind comment. Control variables are used in multiple regression analysis to mitigate the interference of confounding variables in the estimation of causal effects. Reasonable selection of control variables helps to better identify the causal relationship between the explained variables and the core explanatory variables. Therefore, we added 8 control variables in our regression models and introduce what each control variable represents and explain in detail why we control for them in the regression in the part of Control variables (Line 217-254). But the control variables themselves usually do not have a structural explanation, because they are often associated with other unobserved factors, which makes their marginal effect unexplained (Westreich, Greenland, 2013; Keele et al., 2020). As a result, we did not explain the coefficients on the control variables too much in analyzing the regression results. (Liang, Zeger, 1995). We have refined the discussion section from the other side by adding a comparison of the main findings of this paper with previous literature.

References

1. Westreich, D.; Greenland, S. The table 2 fallacy: Presenting and interpreting confounder and modifier coefficients. American journal of epidemiology 2013, 177, 292-298.

2. Keele, L.; Stevenson, R.T.; Elwert, F. The causal interpretation of estimated associations in regression models. Political Science Research and Methods 2020, 8, 1-13.

3. Liang, K.Y.; Zeger, S.L. Inference based on estimating functions in the presence of nuisance parameters. Statistical Science 1995, 10, 158-173.

Sincerely yours,

Kehan Shi

Jinfang Wang

Xiaojin Liu

Xiaoying Zhao

---

## [Decision Letter · Decision Letter 1]

6 Sep 2022

PONE-D-22-18625R1Impact of high-speed rail on tourism in ChinaPLOS ONE

Dear Dr. Liu,

Thank you for submitting your manuscript to PLOS ONE. After careful consideration, we feel that it has merit but does not fully meet PLOS ONE’s publication criteria as it currently stands. Therefore, we invite you to submit a revised version of the manuscript that addresses the points raised during the review process.

We look forward to receiving your revised manuscript.

Kind regards,

Jun Yang

Academic Editor

PLOS ONE

Journal Requirements:

Additional Editor Comments (if provided):

(1) Line 103-114. The research should highlight the differences with the existing research in terms of research object and research scope, so as to further clarify the innovation of selecting 286 cities in China.

(2) Comparison with previous studies should be given, including the advance of the methods and the accuracy of the conclusions. It is essential to claim the novelty of the article.

(3) Can spatial compression be expressed by Figures?

(4) Move the abbreviations either at the beginning or end of the article.

Reviewers' comments:

Reviewer's Responses to Questions

**Comments to the Author**

1. If the authors have adequately addressed your comments raised in a previous round of review and you feel that this manuscript is now acceptable for publication, you may indicate that here to bypass the “Comments to the Author” section, enter your conflict of interest statement in the “Confidential to Editor” section, and submit your "Accept" recommendation.

Reviewer #1: All comments have been addressed

Reviewer #2: All comments have been addressed

2. Is the manuscript technically sound, and do the data support the conclusions?

Reviewer #1: Yes

Reviewer #2: Yes

3. Has the statistical analysis been performed appropriately and rigorously? 

Reviewer #1: Yes

Reviewer #2: Yes

4. Have the authors made all data underlying the findings in their manuscript fully available?

Reviewer #1: Yes

Reviewer #2: Yes

5. Is the manuscript presented in an intelligible fashion and written in standard English?

Reviewer #1: Yes

Reviewer #2: Yes

6. Review Comments to the Author

Reviewer #1: (No Response)

Reviewer #2: (1) Line 103-114. The research should highlight the differences with the existing research in terms of research object and research scope, so as to further clarify the innovation of selecting 286 cities in China.

(2) Comparison with previous studies should be given, including the advance of the methods and the accuracy of the conclusions. It is essential to claim the novelty of the article.

(3) Can spatial compression be expressed by Figures?

(4) Move the abbreviations either at the beginning or end of the article.

7. PLOS authors have the option to publish the peer review history of their article (what does this mean?). If published, this will include your full peer review and any attached files.

Reviewer #1: No

Reviewer #2: No

---

## [Author Response · Author response to Decision Letter 1]

11 Sep 2022

Dear editor:

Thank you for your kind letters of “PLOS ONE Decision: Revision required [PONE-D-22-18625R1]” on 06-Sep-2022, and for the reviewers’ comments concerning our manuscript entitled “Impact of high-speed rail on tourism in China”. These comments are of great help in improving the quality of our papers. We revised the manuscript in accordance with the editor’s and reviewers’ comments, and carefully proof-read the manuscript to minimize typographical, grammatical, and bibliographical errors.

Here below is our description on revision according to the reviewers’ comments. (The review comments are in black, and our response notes are in purple)

Reviewer 2

(1) Line 103-114. The research should highlight the differences with the existing research in terms of research object and research scope, so as to further clarify the innovation of selecting 286 cities in China.

The authors’ Answer: Thank you for your kind comment. As you said, the selection of 286 cities in China is also one of the innovative contributions of this paper. By reviewing the existing literature, we found that most of the literature takes a single HSR line as the research object to analyze its impact on the development of urban tourism, which may lead to the following two problems: first, different HSR lines are built in different regions, and the research conclusions may imply obvious regional characteristics. Second, the additive effect of multiple HSR lines may be wrongly attributed to the effect of the one HSR that is the subject of the study (Line108-111). In this paper, we collated data on all HSR line in China since 2008 and conducted empirical tests on a sample of 286 cities in China to avoid possible regional limitations of the study findings and estimation bias caused by the omission of cross lines (Line 115-118). As suggested, we have added the supplementary explanation for this innovation in the Introduction section.

(2) Comparison with previous studies should be given, including the advance of the methods and the accuracy of the conclusions. It is essential to claim the novelty of the article.

The authors’ Answer: Thank you for your kind comment. In the discussion section, we have compared our results with existing literature and claimed our innovative contribution (Line 534-548). As suggested, we improved this part and refined some content to add to the Introduction section (Line 118-121). First, most of the existing literature focuses on the horizontal effect of high-speed rail on economic growth, ignoring the rate effect, which may lead to an overestimation of the effect of high-speed rail. Our study makes up for this by confirming the existence of a rate effect. However, we do not give a clear answer to the question that the role of high-speed rail may be overestimated. After comparing our estimation results with Zeng and Chen’s [1], we found that the rate effect is indeed lower than the level effect, and we added these in the revised draft. Second, the heterogeneity of the existing literature on high-speed rail is attributed to differences in the socioeconomic characteristics of cities, ignoring the impact of differences in high-speed rail itself. This is not conducive to an accurate understanding of the role of high-speed rail in network operation. Our study breaks this limitation and confirms the existence of the node effect. Third, we analyze the comparative advantages of each city in tourism specialization based on node effect, which provides a new perspective to explain the heterogeneous impact of high-speed rail on tourism development.

(3) Can spatial compression be expressed by Figures?

The authors’ Answer: Thank you for your kind comment. As suggested, we draw a schematic figure of the time-space compression effect of HSR (Fig 1) (Line 140-141), add relevant explanations (Line 131-135) and adjusted the serial number of figures in the article. When the spatial distance between cities is fixed, the opening of HSR shortens the commuting time between cities on the basis of improving the accessibility of cities (Fig 1a). When the commuting time to the destination city is fixed, the opening of HSR makes the spatial structure between cities more compact (Fig 1b).

(4) Move the abbreviations either at the beginning or end of the article.

The authors’ Answer: Thank you for your kind comment. In the last revision of the manuscript, we added a column on the meaning of abbreviations in Table 1 to make the meaning of each abbreviation clearer to the reader (Line 287). But that may still not be intuitive enough. We have carefully considered your comments. In accordance with the principle of "Figures and Tables follow the text", we did not move Table 1, but we also made the following two improvements. First, from Table 2 to Table 6, we explain the abbreviations in the table one by one in the form of table notes (Line 313-315, Line 404-407, Line 440-443, Line 452-455, Line 497-501), such as “ES = Economic scale; FE = Fiscal expenditure; TCD = Transportation convenience degree; IS = Indus-trial structure; SIS = Service industry support; PS = Population size; IL = Income level; OD = Opening degree.”. Second, we made a comparison table of abbreviations (S1 Table) and put it at the end of the article as Supporting information (Line 724-725). The table lists all abbreviations, their corresponding full names and the proxy indicators used.

References

1. Zeng, Y.; Chen, J. The heterogeneous effect of high-speed rails on urban tourism development: An analysis based on the difference-in-differences approach. Tourism Science 2018, 32, 79-92, doi:10.16323/j.cnki.lykx.2018.06.006.

Sincerely yours,

Kehan Shi

Jinfang Wang

Xiaojin Liu

Xiaoying Zhao

---

## [Decision Letter · Decision Letter 2]

19 Sep 2022

PONE-D-22-18625R2Impact of high-speed rail on tourism in ChinaPLOS ONE

Dear Dr. Liu,

Thank you for submitting your manuscript to PLOS ONE. After careful consideration, we feel that it has merit but does not fully meet PLOS ONE’s publication criteria as it currently stands. Therefore, we invite you to submit a revised version of the manuscript that addresses the points raised during the review process.

We look forward to receiving your revised manuscript.

Kind regards,

Jun Yang

Academic Editor

PLOS ONE

Journal Requirements:

Additional Editor Comments (if provided):

Minor Revision

Reviewers' comments:

Reviewer's Responses to Questions

**Comments to the Author**

1. If the authors have adequately addressed your comments raised in a previous round of review and you feel that this manuscript is now acceptable for publication, you may indicate that here to bypass the “Comments to the Author” section, enter your conflict of interest statement in the “Confidential to Editor” section, and submit your "Accept" recommendation.

Reviewer #1: All comments have been addressed

Reviewer #2: (No Response)

2. Is the manuscript technically sound, and do the data support the conclusions?

Reviewer #1: Yes

Reviewer #2: (No Response)

3. Has the statistical analysis been performed appropriately and rigorously? 

Reviewer #1: Yes

Reviewer #2: (No Response)

4. Have the authors made all data underlying the findings in their manuscript fully available?

Reviewer #1: Yes

Reviewer #2: (No Response)

5. Is the manuscript presented in an intelligible fashion and written in standard English?

Reviewer #1: Yes

Reviewer #2: (No Response)

6. Review Comments to the Author

Reviewer #1: (No Response)

Reviewer #2: The document presented is editorially hideous, such as Paragraph spacing(Introduction, Conclusions and discussion), Table expression, etc. It is the authors' responsibility to present their research in a clear and understandable way so that it can be used by all potential interested parties.

7. PLOS authors have the option to publish the peer review history of their article (what does this mean?). If published, this will include your full peer review and any attached files.

Reviewer #1: No

Reviewer #2: No

---

## [Author Response · Author response to Decision Letter 2]

30 Sep 2022

Dear editor:

Thank you for your kind letters of “PLOS ONE Decision: Revision required [PONE-D-22-18625R2]” on 19-Sep-2022, and for the reviewers’ comments concerning our manuscript entitled “Impact of high-speed rail on tourism in China”. These comments have been very helpful in improving the qual-ity of our papers. Changes have been made in response to comments made by the reviewers.

Here below is our description on revision according to the reviewers’ comments. (The review com-ments are in black, and our response notes are in purple)

Reviewer 2

(1) The document presented is editorially hideous, such as Paragraph spacing(Introduction, Conclu-sions and discussion), Table expression, etc. It is the authors' responsibility to present their research in a clear and understandable way so that it can be used by all potential interested parties.

The authors’ Answer: We apologize for any problems with the typesetting of this document. In order to ensure that our manuscript meets PLOS ONE’s style requirements, we modified it according to the PLOS ONE style templates (http://journals.plos.org/plosone/s/file?id=ba62/PLOSOne_formatting_sample_title_authors_affiliations.pdf and http://journals.plos.org/plosone/s/file?id=wjVg/PLOSOne_formatting_sample_main_body.pdf). 

First, we use the double-spaced paragraph format. Second, we set the text-indent to 1 character, except for the first paragraph below the head of each level. Third, we have modified the form of tables in our manuscript according to the PLOS ONE Table Guidelines (https://journals.plos.org/plosone/s/tables) by adding border-right and border-left.

In addition, we resize the affiliation font to 10pt and remove postal codes from the affiliation.

Sincerely yours,

Kehan Shi

Jinfang Wang

Xiaojin Liu

Xiaoying Zhao

---

## [Decision Letter · Decision Letter 3]

6 Oct 2022

Impact of high-speed rail on tourism in China

PONE-D-22-18625R3

Dear Dr. Liu,

We’re pleased to inform you that your manuscript has been judged scientifically suitable for publication and will be formally accepted for publication once it meets all outstanding technical requirements.

Kind regards,

Jun Yang

Academic Editor

PLOS ONE

Additional Editor Comments (optional):

Accept

Reviewers' comments:

Reviewer's Responses to Questions

**Comments to the Author**

1. If the authors have adequately addressed your comments raised in a previous round of review and you feel that this manuscript is now acceptable for publication, you may indicate that here to bypass the “Comments to the Author” section, enter your conflict of interest statement in the “Confidential to Editor” section, and submit your "Accept" recommendation.

Reviewer #1: All comments have been addressed

Reviewer #2: All comments have been addressed

2. Is the manuscript technically sound, and do the data support the conclusions?

Reviewer #1: Yes

Reviewer #2: Yes

3. Has the statistical analysis been performed appropriately and rigorously? 

Reviewer #1: Yes

Reviewer #2: Yes

4. Have the authors made all data underlying the findings in their manuscript fully available?

Reviewer #1: Yes

Reviewer #2: Yes

5. Is the manuscript presented in an intelligible fashion and written in standard English?

Reviewer #1: Yes

Reviewer #2: Yes

6. Review Comments to the Author

Reviewer #1: (No Response)

Reviewer #2: The manuscript has been completely revised according to the comments and meets the requirements for publication. I agree to publish it.

7. PLOS authors have the option to publish the peer review history of their article (what does this mean?). If published, this will include your full peer review and any attached files.

Reviewer #1: No

Reviewer #2: No
